# DRMF: Degradation-Robust Multi-Modal Image Fusion via Composable Diffusion Prior

Submission Id: 2367

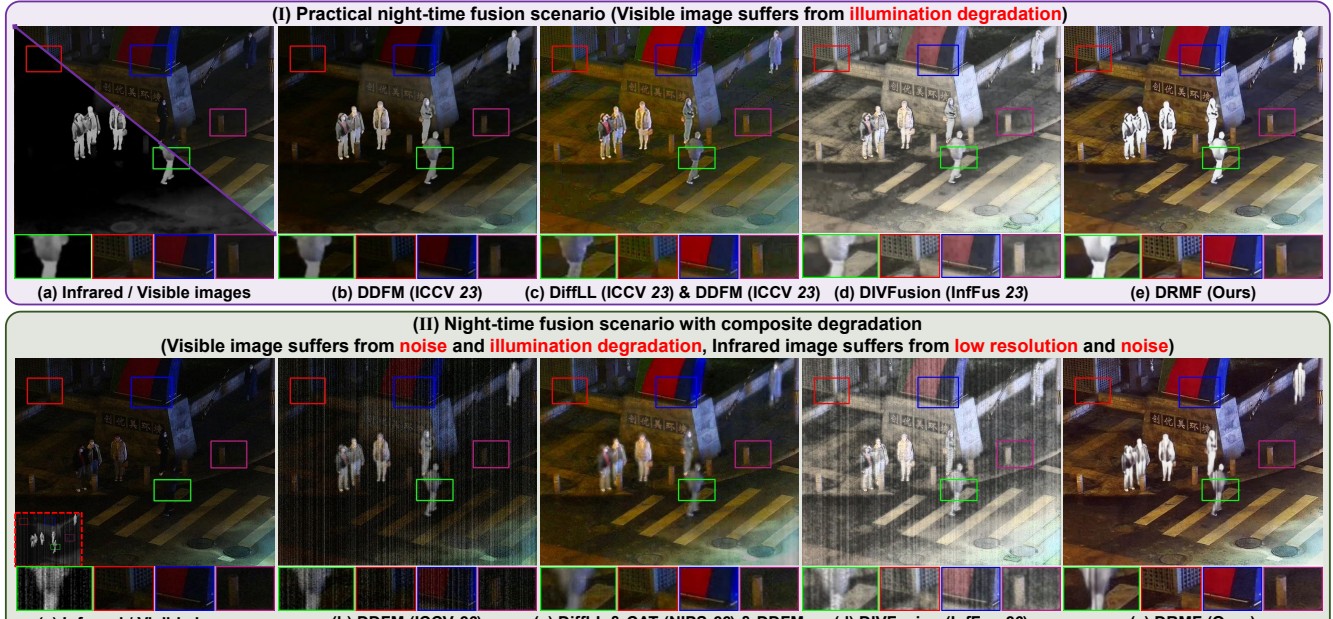

**Figure 1: Fusion schematic in challenging scenarios for MMIF tasks. DDFM [64] and DiffLL [12] are Diffusion-based methods for image fusion and low-light image enhancement. DIVFusion [40] is an illumination-robust image fusion method, and CAT [2] is a SOTA universal image restoration method for preparatory denoising and super-resolution.**

## ABSTRACT

Existing multi-modal image fusion algorithms are typically designed for high-quality images and fail to tackle degradation (*e.g.*, low light, low resolution, and noise), which restricts image fusion from unleashing the potential in practice. In this work, we present **D**egradation-**R**obust **M**ulti-modality image **F**usion (**DRMF**), leveraging the powerful generative properties of diffusion models to counteract various degradations during image fusion. Our critical insight is that generative diffusion models driven by different modalities and degradation are inherently complementary during the denoising process. Specifically, we pre-train multiple degradation-robust conditional diffusion models for different modalities to handle degradations. Subsequently, the diffusion priori combination module is devised to integrate generative priors from pre-trained uni-modal models, enabling effective multi-modal image fusion. Extensive experiments demonstrate that DRMF excels in infrared-visible and medical image fusion, even under complex degradations.

## CCS CONCEPTS

• **Computing methodologies → Computer vision**.

## KEYWORDS

Multi-modal image fusion, image enhancement, diffusion model

## 1 INTRODUCTION

Multi-modal image fusion (MMIF) is an essential image enhancement technique that aggregates significant information from diverse sensors or modalities for a comprehensive scene representation [13]. Infrared-visible image fusion (IVIF) and medical image fusion (MIF) are the most representative MMIF tasks. IVIF aims to merge essential thermal radiation information from infrared (IR) images and rich texture from visible (VI) images. The fusion results can overcome the limitations of IR images suffering from noise and low resolution as well as VI images affected by illumination and camouflage [60]. MIF seeks to integrate functional and metabolic information in functional images with structural and anatomical

information in structural images into a single image, aiding in subsequent disease diagnosis and treatment [7]. Sufficient information aggregation and pleasing visual outcomes enable MMIF to be widely applicable across various domains, such as intelligent medical service [67], nighttime assisted driving [1], object detection [10], and semantic segmentation [58].

In recent years, MMIF has gained increasing attention, and numerous algorithms have been developed to achieve greater visual appeal. Owing to powerful feature extraction and representation abilities, auto-encoder (AE)-based [16, 17, 21, 63], convolutional neural network (CNN)-based [27, 40, 51, 62], generative adversarial network (GAN)-based [20, 28, 29] and Transformer-based [26, 44, 57] frameworks dominate the MMIF field. In addition, the diffusion model [8, 35] with strong distribution modeling and generative abilities shows great potential for MMIF [55, 64]. DDFM [64] migrates the generative prior of the pre-trained large-scale natural image diffusion model to MMIF using the score matching technique [38]. However, the implementation of score matching involves tedious design and requires re-optimizing an objective function for each specific scenario, making it time-consuming. Moreover, the distribution variation between natural and multi-modal images limits the performance of pre-trained diffusion models in MMIF.

Additionally, most existing fusion methods are tailored for normal scenarios. As illustrated in Figure 1 (I), the VI image in a realistic night scene experiences severe illumination degradation. Consequently, even the advanced diffusion model-based fusion method (*i.e.*, DDFM) struggles to represent the scene information effectively. Several works attempt to simultaneously enhance and aggregate complementary information in challenging scenarios [40, 48]. DIVFusion [40] first leverages the Retinex theory to extract reflectance-related and illumination-related features from VI images. Then, it only integrates reflectance-related features in the subsequent fusion stage, mitigating the negative impact of illumination degradation. Nevertheless, since the illumination component is completely dropped, the fusion result of DIVFusion, shown in Figure 1 (I (d)), appears unnatural and suffers from noticeable color distortion. Note that source images in challenging scenarios may be affected by multiple degradations, including noise, low contrast, low resolution, low light, *etc.* As illustrated in Figure 1(II), existing fusion methods struggle to effectively address composite degradation, leading to fusion results that exhibit noise and blurriness, which greatly restrict the practical application of MMIF. It is worth mentioning that some preprocessing techniques (*e.g.* DiffLL [12] for low-light image enhancement and CAT [2] for denoising as well as super-resolution) can be utilized to pre-enhance source images. Nevertheless, the poor coupling between multiple cascading tasks leads to error amplification and thus synthesizes unsatisfactory fusion results, as presented in Figure 1 (I (c)) and (II (c)).

To overcome the above challenges, we propose a degradation-robust fusion method based on composable diffusion prior, termed DRMF, which is a novel diffusion-based paradigm for MMIF. DRMF fully leverages the inherent complementary potential of diffusion models driven by different modalities and degradations to mitigate various degradations. We begin by pre-training multiple degradation-robust conditional diffusion models, which are conditioned on degraded source images to estimate the distribution of corresponding

high-quality images. Since generative diffusion priors are characterized by Gaussian noise, the priors of various diffusion models can be combined in the noise space. Specifically, during the sampling process, given the fusion sample $x_t^f$ at $t$-step, the diffusion model $\epsilon_\theta^i$ driven by $i$-modality is employed to estimate the noise (*i.e.*, generative prior) $n_t^i$ to approximate high-quality revision of $i$-modality. Then, the diffusion priori combination module estimates the combination weights of different modality-driven diffusion priors $n_t^i$ and produces the next sample $x_{t-1}^f$. Gradually, DRMF can yield a high-quality fused image conditioned on degraded source images. As shown in Figure 1, DRMF effectively addresses various complex degradations and generates impressive fusion results that emphasize significant targets and vividly portray scene details. To sum up, our major contributions are threefold:

- We propose a novel diffusion model-based framework for MMIF, which harnesses the powerful generative properties of diffusion models to mitigate various complex degradations in source images.
- We develop a diffusion prior combination module to aggregate generative diffusion priors from different modalities, effectively leveraging the complementary nature of diffusion models driven by various modalities and degradations.
- Extensive experiments demonstrate the superiority of our method in the IVIF and MIF tasks, especially when source images are afflicted by composite degradations.

## 2 RELATED WORK

**Multi-modal image fusion.** The booming development of deep learning has energized the field of MMIF. AE [16, 65], CNN [51, 62], GAN [20, 29], and Transformer [26, 44] are the major network architectures used in deep learning-based MMIF methods. In response to the practical demands of subsequent high-level vision tasks, the image fusion community has proposed numerous semantic-driven approaches [20, 22, 41, 43], which constrain the fusion network to retain richer semantic cues. Additionally, RFNet [53], UMF-CMGR [47], SuperFusion [39], and MURF [54] jointly model image registration and fusion tasks, correcting the parallax and aberration in actual shooting. Furthermore, some universal methods [18, 26, 52, 56] uniformly fulfill various image fusion tasks. Specifically, realizing that source images are usually affected by degradation, Tang *et al.* [40] and Wang *et al.* [48] devised illumination-robust fusion approaches, which take into account low-light image enhancement and information fusion. However, these methods are specifically designed for illumination degradation and fail to address other common degradations, such as noise and low resolution. In this work, we utilize the diffusion models with powerful generative abilities to tackle the challenges of composite degradation.

**Diffusion model.** Recently, the diffusion model [8] has gained widespread attention within the computer vision community, which samples high-quality images from the estimated target distribution. Benefiting from their powerful generative capabilities, diffusion models have found extensive applications in various low-level visual tasks, such as text-to-image generation [3, 35], image manipulation [15], super-resolution [36], de-raining [30], de-blurring [50], and low-light image enhancement [5], consistently delivering remarkable results. In addition, composable diffusion

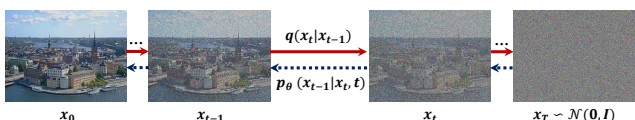

**Figure 2: Forward and reverse processes of diffusion models.**

models [9, 15, 23] are proposed to exploit multi-modality conditions for controlling the generation process more accurately.

The diffusion model is also introduced to the field of MMIF. Dif-Fusion [55] for the first time employs the denoising network of the diffusion model as a potent feature extractor, while the whole image fusion process is detached from the diffusion model. DDFM [64] combines the score matching and a pre-trained unconditional diffusion model to transfer the generative prior learned from natural images to multi-modal images for image fusion. Notably, existing diffusion model-based methods are not specifically designed for the challenges in MMIF. As a result, the remarkable generative abilities of diffusion models are not unleashed. In this work, we design task-specific conditional diffusion models tailored to cope with various degradations in multi-modal images. Then, we propose a diffusion prior combination module to achieve image fusion using generative priors driven by different modalities and degradations.

## 3 METHOD

Given complementary degraded source images $\{x_c^i\}$ where $i \in \{ir, vi, ct, mri\}$, our method starts from random Gaussian noises $x_T$ and generates high-quality fused images $x^f$ by step-by-step denoising conditional on $\{x_c^i\}$. We devise a diffusion prior combination module to combine the noises $\{n_t^i\}$ from multiple diffusion models $\{\epsilon_\theta^i\}$ and infer the next fusion sample during the denoising process. In particular, the estimated noises characterize the degradation-robust diffusion priors driven by different modalities and degradations, thereby we can achieve multi-modality information aggregation. Our sampling process adheres to the standard conditional diffusion model. Therefore, in the following we present the typical diffusion model, degradation-robust conditional diffusion models, and diffusion prior combination modules in sequence.

### 3.1 Diffusion Preliminary

**Denoising diffusion probabilistic model** [8] is a classical generative model that utilizes a Markov chain to transform complex data distributions into a simple Gaussian noise distribution. It then recovers the desired data distribution from the noise distribution through step-by-step denoising. As shown in Figure 2, in the forward process, the given data distribution $x_0 \sim p(x_0)$ is progressively corrupted through a pre-defined Marchkov chain. This is accomplished by gradually introducing Gaussian noise, and the pre-defined Markov chain satisfies the following distribution:

$$q(x_t|x_{t-1}) = \mathcal{N}(x_t; \sqrt{\alpha_t}x_{t-1}, \beta_t I), \tag{1}$$

where $\beta_t$ governs the noise variance added at $t$-step and $\alpha_t = 1 - \beta_t$. Thus the marginal distribution is derived as:

$$q(x_t|x_0) = \mathcal{N}(x_t; \sqrt{\bar{\alpha}_t}x_0, \bar{\beta}_t I), \tag{2}$$

where $\bar{\alpha}_t = \Pi_{i=1}^t \alpha_i$ and $\bar{\beta}_t = 1 - \bar{\alpha}_t$. As $t$ approaches a large value $T$, $\bar{\alpha}_T$ tends to 0 and $q(x_T|x_0)$ approximates the normal distribution $\mathcal{N}(0, I)$, the forward process is finished.

The reverse process starts with a Gaussian noise $x_T \sim \mathcal{N}(0, I)$ and progressively denoises to generate a clean image through a Markov chain, defined as:

$$p_\theta(x_{t-1}|x_t) = \mathcal{N}(x_{t-1}; \mu_\theta(x_t, t), \sigma_t^2 I). \tag{3}$$

As presented in [8], the mean $\mu_\theta(x_t, t)$ is calculated based on estimated results of the noise estimation network $\epsilon_\theta$ with parameters $\theta$, and the variance $\sigma_t^2$ is a time-dependent constant. Specifically, the above processes are formulated as:

$$\mu_\theta(x_t, t) = \frac{1}{\sqrt{\alpha_t}}(x_t - \frac{\beta_t}{\sqrt{1-\bar{\alpha}_t}}\epsilon_\theta(x_t, t)),$$
$$\sigma_t^2 = \frac{(1-\bar{\alpha}_{t-1})}{(1-\bar{\alpha}_t)}\beta_t. \tag{4}$$

Ultimately, the optimization objective is defined as:

$$\mathbb{E}_{x_0, n_t, t}\left[\|n_t - \epsilon_\theta(\sqrt{\bar{\alpha}_t}x_0 + \sqrt{1-\bar{\alpha}_t}n_t, t)\|^2\right], \tag{5}$$

where $n_t$ is sampled from a standard Gaussian noise.

**Denoising diffusion implicit model** [37] presents an accelerated deterministic sampling manner for pre-trained diffusion models, where the forward process follows a non-Markovian process. The marginal distribution $q(x_t|x_0)$ still obeys Eq. (2), and the mean in Eq. (3) is re-formulated as:

$$\mu_\theta(x_t, t) = \sqrt{\bar{\alpha}_{t-1}}\hat{x}_0 + \sqrt{1-\bar{\alpha}_{t-1}-\lambda_t^2}\epsilon_\theta(x_t, t), \tag{6}$$

where $\hat{x}_0 = \frac{x_t - \sqrt{1-\bar{\alpha}_t}\epsilon_\theta(x_t, t)}{\sqrt{\bar{\alpha}_t}}$ indicates the estimated $x_0$ at the current step, and $\lambda_t$ is a tunable hyper-parameter controlling the variance of the marginal distribution. Thus, $\sigma_t^2$ in Eq. (3) is set to $\lambda_t^2$.

### 3.2 Degradation-robust Conditional Diffusion

Conditional diffusion models have been shining in the field of image generation [3, 35], enhancement [5], and degraded image restoration [30, 36, 50]. However, diffusion models in MMIF are primarily employed to extract features [55] and provide routine generative priors of natural images [64] without tackling degradations. In this work, we first present degradation-robust conditional diffusion models (DRCDMs) to restore degraded source images. As shown in Figure 3, given the degraded image $x_c^i$ and the high-quality versions $x_0^i$, the forward process in the training stage follows Eq. (1), i.e., $q(x_t^i|x_{t-1}^i) = \mathcal{N}(x_t^i; \sqrt{\alpha_t}x_{t-1}^i, \beta_t I)$. The reverse process is defined as $p_\theta(x_{t-1}^i|x_t^i, x_c^i) = \mathcal{N}(x_{t-1}^i; \mu_\theta(x_t^i, x_c^i, t), \sigma_t^2 I)$, where the posterior is further conditioned on degraded image $x_c^i$ as:

$$\mu_\theta(x_t^i, x_c^i, t) = \frac{1}{\sqrt{\alpha_t}}(x_t^i - \frac{\beta_t}{\sqrt{1-\bar{\alpha}_t}}\epsilon_\theta^i(x_t^i, x_c^i, t)). \tag{7}$$

Specifically, the intermediate sample $x_t^i$ is concatenated with the degraded image $x_c^i$ and fed into the noise estimation network $\epsilon_\theta^i$, where the degraded image plays a crucial role in providing valuable semantic and structural information. Furthermore, the optimization objective is modified as:

$$\mathbb{E}_{x_0^i, x_c^i, n_t, t}\left[\|n_t - \epsilon_\theta^i(\sqrt{\bar{\alpha}_t}x_0^i + \sqrt{1-\bar{\alpha}_t}n_t, x_c^i, t)\|^2\right]. \tag{8}$$

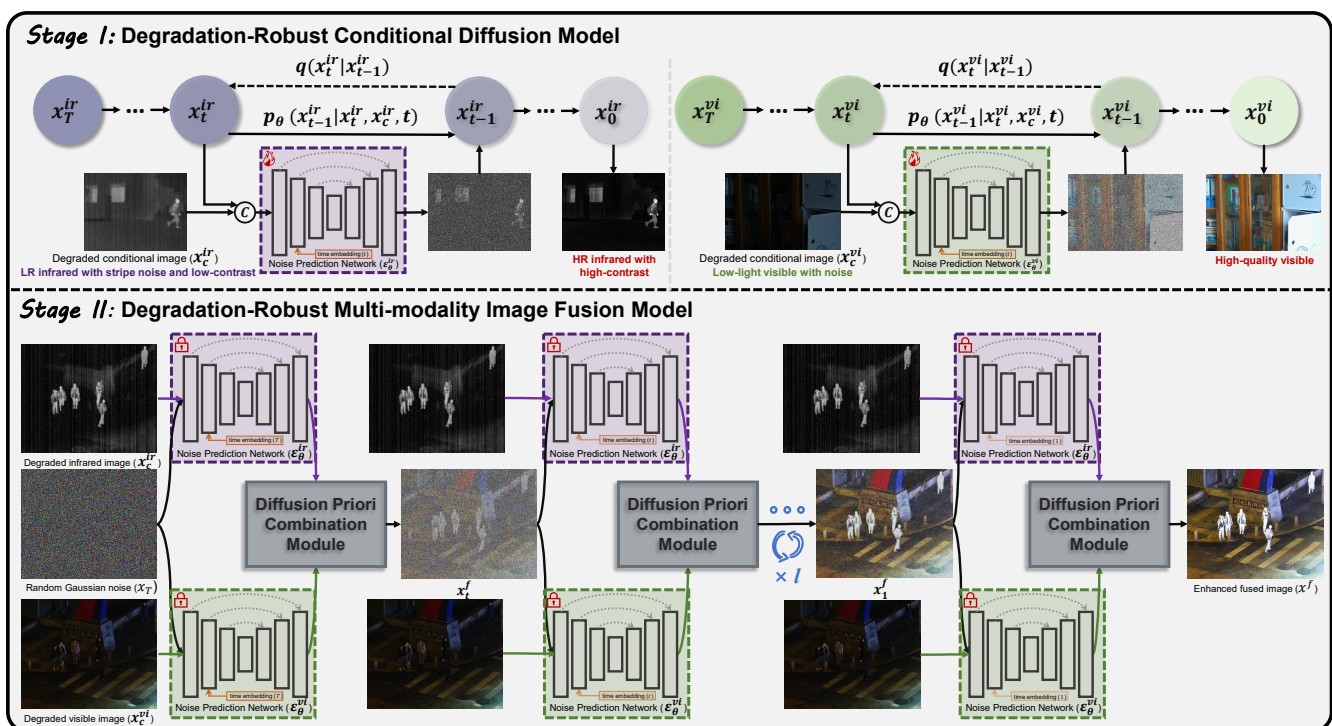

**Figure 3: The framework of our DRMF (IVIF as an example). Stage I presents the pipeline of the DRCDMs. Stage II indicates the procedure of directly generating fusion results with the pre-trained degradation-robust diffusion prior. Given complementary degraded images $\{x_c^i\}$ and arbitrary Gaussian noise $x_T^f$, DRMF first predicts the degradation-robust generative prior (*i.e.*, noise) $\{n_t^i\}$ using noise estimation networks $\{\epsilon_\theta^i\}$. Then, the generative priors provided by various modalities are aggregated with the DPCM to infer the subsequent fusion sample $x_t^f$ until the final high-quality fused image $x^f$ is generated.**

During the inference stage, high-quality source images are generated via iterative denoising, starting from a standard Gaussian noise and guided by degraded conditional images. Note that in our solution, DRCDMs not only recover high-quality images from degraded ones but also offer degradation-robust generative diffusion priors, which are employed to yield high-quality fused images directly.

### 3.3 Diffusion Prior Combination

After pre-training, DRCDMs have learned the prior distributions of high-quality uni-modal images. The priors are determined by noises predicted at each sampling step. Related works [9, 15, 23] confirm that combining noises from multiple diffusion models can generate a result containing corresponding prior distributions. Therefore, to directly generate a fused image that carries characteristics from high-quality uni-modal priors, we devise a diffusion prior combination module (DPCM) to flexibly merge complementary noises. In detail, as shown in Figure 3, we invert the image from random Gaussian noise with pre-trained DRCDMs and update the mean during the DDIM sampling process according to the rule:

$$\mu(x_t^f, x_c, t) = \sqrt{\bar{\alpha}_{t-1}} \sum_i \gamma_t^i \hat{x}_{0_t}^i + \sqrt{1 - \bar{\alpha}_{t-1} - \lambda_t^2} \sum_i \gamma_t^i \epsilon_\theta^i(x_t^f, x_c^i, t),$$
$$(9)$$

where $\hat{x}_{0_t}^i = \frac{x_t^f - \sqrt{1-\bar{\alpha}_t} \epsilon_\theta^i(x_t^f, x_c^i, t)}{\sqrt{\bar{\alpha}_t}}$ denotes the estimated sample associated with $i$-modality conditioned on $x_c^i$ from the fusion sample at $t$-step. $\{\gamma_t^i\}_{t=1}^T$ is the sequence weights for each generative diffusion prior satisfying $\sum_i \gamma_t^i = 1$.

In this way, the goal of DPCM is to estimate sequence weights for each modality. The learnable weights need to take into account several considerations. They 1) should measure the complementary properties of different modalities, 2) are expected to be learned from high-quality images, 3) are modulated by timestep $t$, and 4) are guided by the weight from the previous step. Therefore, the process of generating weights with a U-Net $\varphi$ is defined as:

$$\hat{\gamma}_t^i = \varphi(x_t^f, \hat{x}_{0_t}^i, \gamma_{t+1}^i, t). \qquad (10)$$

The current fusion sample $x_t^f$, the predicted high-quality sample $\hat{x}_{0_t}^i$ for $i$-modality, and the weight $\gamma_{t+1}^i$ from the previous step are concatenated and fed into the U-Net. To normalize weights of different modalities, we perform softmax across all weights $\{\hat{\gamma}_t^i\}$ to obtain the final combination weight $\{\gamma_t^i\}$ by $\gamma_t^i = \frac{\exp(\hat{\gamma}_t^i)}{\sum_j \exp(\hat{\gamma}_t^j)}$. Specifically, in the fusion task involving two complementary modalities, one weight $\gamma_t^i$ is estimated, and the weight for the other modality can be directly derived as $1 - \gamma_t^i$.

---

**Algorithm 1:** Training Algorithm of DPCM

**Input:** Pre-trained diffusion models $\{\epsilon_\theta^i\}$, degraded images $\{x_c^i\}$, and their estimated high-quality versions $\{\hat{x}_0^i\}$

1 **repeat**

2  Initializing sample $x_T^f \sim \mathcal{N}(0, I)$;

3  **for** $t = T - 1, \cdots, 1$ **do**

4   **for** $i$ in $\{ir, vi\}$ **do**

5    $\hat{x}_{0_t}^i \leftarrow \dfrac{x_t^f - \sqrt{1-\bar{\alpha}_t}\epsilon_\theta^i(x_t^f, x_c^i, t)}{\sqrt{\bar{\alpha}_t}}$;

6    $\hat{\gamma}_t^i \leftarrow \varphi(x_t^f, \hat{x}_{0_t}^i, \gamma_{t+1}^i, t)$;

7   **end**

8   $\gamma_t^i \leftarrow \dfrac{\exp(\hat{\gamma}_t^i)}{\sum_j \exp(\hat{\gamma}_t^j)}$;

9   $\hat{x}_{0_t}^f \leftarrow \dfrac{x_t^f - \sqrt{1-\bar{\alpha}_t}\sum_i \gamma_t^i \epsilon_\theta^i(x_t^f, x_c^i, t)}{\sqrt{\bar{\alpha}_t}}$;

10   Updating $x_{t-1}^f$ by Eq. (9);

11  **end**

12  Calculating $\mathcal{L}$ by Eq. (11), Eq. (12) and Eq. (13);

13  Taking gradient descent step on $\nabla_\varphi \mathcal{L}$;

14 **until** *converged*;

---

We also devise corresponding losses to optimize the weight prediction network $\varphi$. The fusion loss is defined as:

$$\mathcal{L}_f = \sum_{i,t}\left(\sqrt{\alpha_t}\|\hat{x}_{0_t}^i - \max_i(\{\hat{x}_0^i\})\|_1 + \sqrt{\alpha_t}\|\nabla\hat{x}_{0_t}^f - \max_i(\{\nabla\hat{x}_0^i\})\|_1\right),$$
(11)

where $\hat{x}_{0_t}^f = \dfrac{x_t^f - \sqrt{1-\bar{\alpha}_t}\sum_i \gamma_t^i \epsilon_\theta^i(x_t^f, x_c^i, t)}{\sqrt{\bar{\alpha}_t}}$ denotes a clean fused image predicted at $t$-step and $\hat{x}_0^i$ is the high-quality source image restored by the pre-trained DRCDM. The $\max(\cdot)$ means the maximum aggregation strategy and $\nabla$ represents the gradient operator [41, 43]. As the denoising proceeds (*i.e.*, as $t$ decreases), the estimated fusion results get closer to the desired ones, thereby we weight the fusion loss with time-dependent coefficient $\sqrt{\alpha_t}$. In addition, we introduce a regularization term to constraint the smoothness of the combination weights $\gamma_t^i$, which is defined as:

$$\mathcal{L}_{re} = \sum_{i,t}\|\nabla\gamma_t^i\|_1.$$
(12)

The full objective function for $\varphi$ is a weighted sum of $\mathcal{L}_f$ and $\mathcal{L}_{re}$:

$$\mathcal{L} = \mathcal{L}_f + \lambda_{re} \cdot \mathcal{L}_{re}.$$
(13)

In particular, the IVIF task is expected to retain salient objects in IR images. To achieve this, we design an additional mask-guided loss, utilizing the salient target mask $m$ to guide $\gamma_t^{ir}$ learning, as defined:

$$\mathcal{L}_m = \sum_t -(m\log(\gamma_t^{ir}) + (1-m)\log(1-\gamma_t^{ir})).$$
(14)

The training procedure of DPCM is summarized in Algorithm 1, which uses IVIF as an example and involves only fusion and regularization losses to ensure generality. During the inference stage, we employ DDIM [37] fast sampling and update $x^f$ at each step according to Eq. (9), iteratively generating high-quality fused images.

# 4 EXPERIMENTS

## 4.1 Implementation Details

We train multiple DRCDMs separately. For IVIF, we initially train DRCDMs for VI and IR images on the LOL [49] and MSRS [42] datasets, respectively. LOL consists of paired low- and norm-light images. We down-sample the low-contrast IR images with down-sampling factor of $1/4$ and add streak noise with $\sigma^2 = 15$ to simulate degraded images. The high-quality versions are provided by MSRS. Subsequently, the illumination-robust CDM pre-trained on LOL is used to enhance the VI images in MSRS. Finally, we employ enhanced VI and high-quality IR images from MSRS to train the DPCM, and the salient target mask $m$ is in Eq. (14) converted from segmentation labels in MSRS. For MIF, we introduce degradation in the form of Gaussian noise (with $\sigma^2 = 25$) to MRI images and down-sample CT images using a down-sampling factor of $1/4$.

During DRCDM training, the batch size is set at 64, the learning rate is set at $2 \times 10^{-5}$, the training iteration is set at $2 \times 10^6$, and all images are randomly cropped into $64 \times 64$. We introduce the attention mechanism at $1/4$ resolution for computational efficiency, and other settings adhere to the default settings of diffusion models [30]. When training the DPCM, we employ DDIM deterministic sampling with $\lambda_t = 0$ for generating fused images and set the sampling step to 5. The batch size and training iteration are adjusted to 1 and $1 \times 10^5$, and source images are randomly cropped to $128 \times 128$.

We first evaluate the fusion performance of the proposed scheme in practical scenarios on LLVIP [11], MSRS, and Harvard [45] datasets, which contain 216, 361, and 20 test images, respectively. Then, we utilize degraded images in these datasets to demonstrate the superiority of DRMF in solving composite degradation for MMIF. Moreover, the test set of LOL with 15 image pairs is used to validate the strong performance of DRCDMs in tackling degradation.

## 4.2 Image Fusion in Practical Scenarios

We first consider practical scenarios, where all source images are captured directly by sensors. DRMF is compared with 8 SOTA methods, *i.e.*, DeFusion [18], TarDAL [20], U2Fusion [52], DIVFusion [40], LRRNet [17], MURF [54], PAIF [25] and DDFM [64]. Visual comparisons for IVIF and MIF tasks are shown in Figures 4 and 5. In nighttime IVIF scenarios, DRMF effectively lights up information in the dark and enhances the contrast of IR images. Our results thus provide bright backgrounds and significant targets. DIVFusion also lights up VI images, but it suffers from overexposure and color saturation reduction. Other approaches struggle to mine valuable information in the dark or even diminish salient targets. As in Figure 5, only DRMF completely integrates functional and structural information in CT and MRI images, benefiting from the adaptive aggregation ability of DPCM. Quantitative results using MI [32], SF [4], SD [33], and VIF [6] are displayed in Table 1. A SOTA low-light enhancement (LLIE) method (*i.e,* DiffLL [12]) is employed to enhance low-light VI images for comparing fusion performance fairly. Orig. denotes the practical fusion results, whereas En. means fusion results with LLIE for VI images. Achieving the best SD and VIF means our method has optimal contrast and visual effects. The superior MI suggests that DRMF effectively transfers valuable information from source images to fusion results. The comparable SF shows that our fusion results integrate abundant textures.

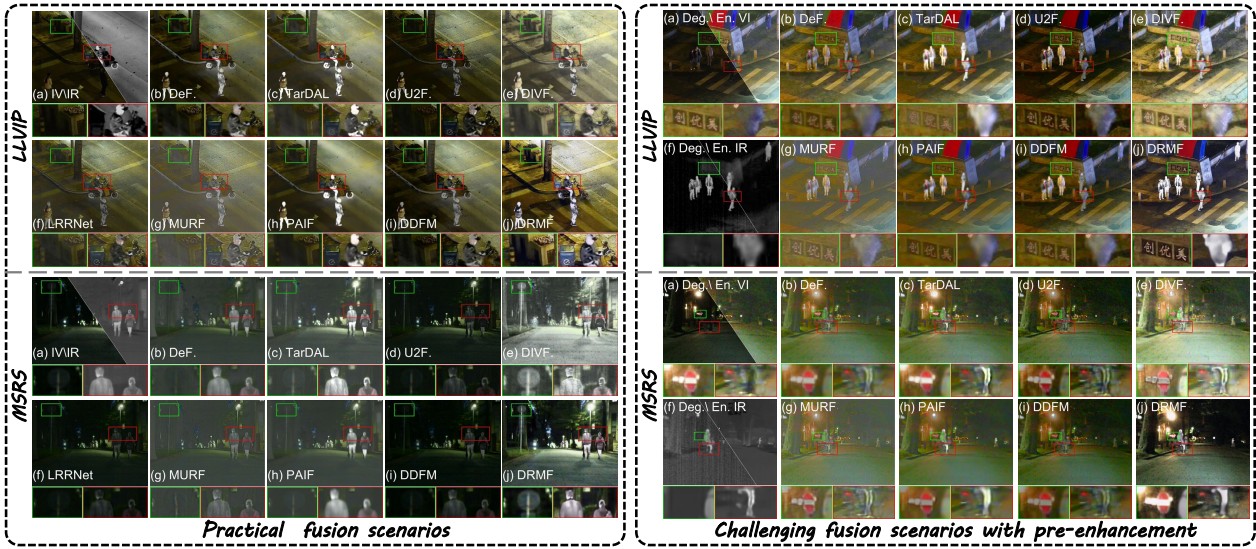

Figure 4: Visual comparison of DRMF with state-of-the-art approaches on practical and challenging fusion scenarios for IVIF.

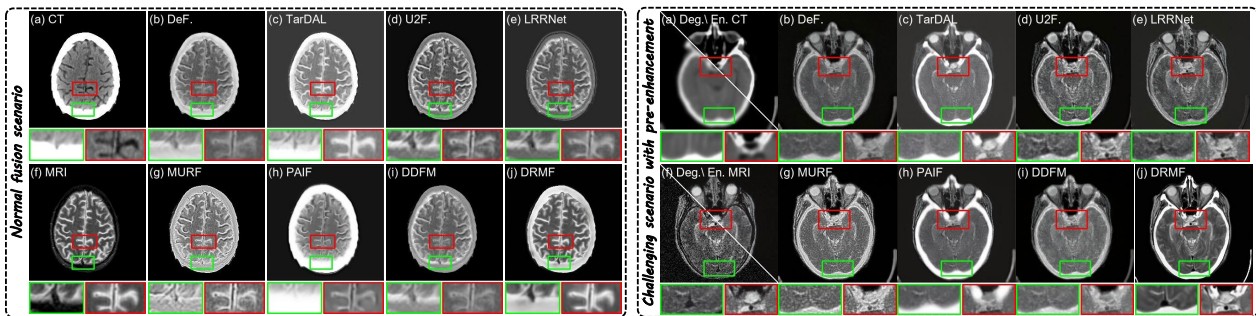

Figure 5: Visual comparison of our DRMF with state-of-the-art approaches on normal and challenging scenarios for MIF.

Table 1: Quantitative comparison of DRMF with state-of-the-art methods on IVIF and MIF tasks in practical fusion scenarios. The best and second results are highlighted in red and *blue*, respectively.

| Method | LLVIP | | | | | | | | MSRS | | | | | | | | CT-MRI | | | |
|---|---|---|---|---|---|---|---|---|---|---|---|---|---|---|---|---|---|---|---|---|
| | MI | | SF | | SD | | VIF | | MI | | SF | | SD | | VIF | | MI | SF | SD | VIF |
| | Orig. | En. | Orig. | En. | Orig. | En. | Orig. | En. | Orig. | En. | Orig. | En. | Orig. | En. | Orig. | En. | Orig. | Orig. | Orig. | Orig. |
| DeF. [18] | *2.98* | 2.15 | 13.97 | 12.81 | 41.09 | 34.72 | 0.60 | 0.47 | 2.71 | 3.00 | 5.95 | 7.00 | 24.10 | 27.89 | 0.67 | 0.62 | 2.69 | 19.59 | 76.22 | 0.45 |
| TarDAL [20] | 2.96 | *2.52* | *17.04* | 17.81 | 50.28 | 51.26 | 0.56 | 0.48 | *3.02* | 3.15 | 10.01 | 11.66 | 36.73 | 41.07 | 0.74 | 0.66 | 2.75 | 20.25 | 67.29 | 0.43 |
| U2F. [52] | 2.11 | 1.91 | 16.94 | *19.71* | 35.34 | 42.81 | 0.52 | 0.52 | 2.77 | 2.70 | 10.50 | **13.53** | 33.44 | 39.07 | 0.75 | 0.73 | 2.38 | 20.82 | 58.74 | 0.39 |
| DIVF. [40] | 2.07 | 2.01 | 16.12 | 16.84 | *53.16* | *53.33* | *0.68* | *0.50* | 2.77 | 2.56 | **12.21** | 14.10 | *53.58* | *53.94* | *0.88* | 0.64 | 2.29 | 14.74 | 55.56 | 0.31 |
| LRRNet [17] | 2.05 | 2.19 | 15.74 | 16.39 | 28.97 | 35.09 | 0.42 | 0.48 | 3.01 | *3.35* | 8.48 | 10.47 | 32.04 | 38.85 | 0.66 | 0.68 | 2.31 | 15.23 | 42.56 | 0.33 |
| MURF [54] | 1.92 | 1.45 | 14.37 | 14.81 | 22.04 | 22.77 | 0.33 | 0.30 | 1.87 | 1.72 | 9.75 | 11.55 | 19.42 | 21.17 | 0.48 | 0.42 | 2.41 | **38.01** | 81.26 | 0.37 |
| PAIF [25] | 2.63 | 2.38 | 14.63 | 12.40 | 46.41 | 40.67 | 0.42 | 0.40 | 2.96 | 3.00 | 7.26 | 7.53 | 24.76 | 24.74 | 0.49 | 0.50 | 2.53 | 18.79 | *86.84* | 0.29 |
| DDFM [64] | 2.52 | 2.01 | 16.73 | 17.04 | 37.86 | 38.22 | *0.59* | *0.51* | 2.75 | 3.00 | 7.46 | *9.80* | 29.24 | 35.43 | 0.79 | *0.76* | **3.23** | 18.11 | 65.44 | *0.47* |
| DRMF (Ours) | **3.82** | **3.82** | **25.83** | **25.83** | **54.02** | **54.02** | **0.93** | **0.93** | **4.58** | **4.58** | *12.05* | *12.05* | **54.26** | **54.26** | **0.97** | **0.97** | *2.81* | *26.42* | **87.74** | **0.54** |

## 4.3 Image Fusion in Challenging Scenarios

Figures 4 and 5 also show visual comparisons of various fusion methods in challenging scenarios with composite degradation. Specifically, VI images experience noise and low light, while IR images endure low contrast, low resolution, and streak noise. CT and MRI

images suffer from low resolution and noise, respectively. These degradations are typical challenges in real-life shoots. For fair comparisons, we employ DiffLL to pre-enhance VI images, and fine-tune CAT [2] to denoise VI, IR, and MRI images. CAT is also fine-tuned to super-resolution IR and CT images. As shown in Figures 4 and 5, our

**Table 2: Quantitative comparison of DRMF with state-of-the-art methods on IVIF and MIF tasks in challenging scenarios.**

| Method | LLVIP MI | | LLVIP SF | | LLVIP SD | | LLVIP VIF | | MSRS MI | | MSRS SF | | MSRS SD | | MSRS VIF | | CT-MRI MI | | CT-MRI SF | | CT-MRI SD | | CT-MRI VIF | |
|---|---|---|---|---|---|---|---|---|---|---|---|---|---|---|---|---|---|---|---|---|---|---|---|---|
| | Orig. | En. | Orig. | En. | Orig. | En. | Orig. | En. | Orig. | En. | Orig. | En. | Orig. | En. | Orig. | En. | Orig. | En. | Orig. | En. | Orig. | En. | Orig. | En. |
| DeF. [18] | 2.84 | 2.36 | 11.49 | 8.82 | 39.62 | 33.57 | 0.56 | 0.53 | 2.22 | 3.23 | 6.57 | 6.52 | 24.41 | 27.53 | 0.54 | 0.69 | 2.73 | 3.41 | 19.67 | 14.24 | 65.59 | 71.19 | 0.36 | 0.52 |
| TarDAL [20] | 2.61 | 2.70 | 17.60 | 14.41 | 49.32 | 49.38 | 0.48 | 0.56 | 2.34 | 3.36 | 13.47 | 10.92 | 36.77 | 40.27 | 0.50 | 0.73 | 3.04 | 3.64 | 24.70 | 18.78 | 68.81 | 67.53 | 0.39 | 0.46 |
| U2F. [52] | 1.93 | 2.01 | 16.35 | 15.22 | 36.79 | 45.42 | 0.43 | 0.60 | 2.03 | 2.88 | 12.39 | 12.29 | 34.64 | 38.12 | 0.49 | 0.82 | 2.74 | 3.12 | 25.72 | 17.38 | 57.73 | 60.54 | 0.32 | 0.40 |
| DIVF. [40] | 1.78 | 2.10 | 16.41 | 15.23 | 53.04 | 53.38 | 0.48 | 0.60 | 1.97 | 2.79 | 15.36 | 13.22 | 53.33 | 53.95 | 0.48 | 0.76 | 2.46 | 3.15 | 10.84 | 14.07 | 55.97 | 55.16 | 0.25 | 0.38 |
| LRRNet [17] | 1.94 | 2.36 | 15.83 | 14.20 | 31.13 | 41.28 | 0.39 | 0.61 | 2.18 | 3.59 | 11.70 | 10.18 | 33.18 | 39.78 | 0.49 | 0.76 | 2.76 | 3.29 | 32.79 | 14.96 | 40.91 | 46.11 | 0.37 | 0.38 |
| MURF [54] | 1.75 | 1.58 | 12.93 | 11.56 | 22.26 | 23.16 | 0.31 | 0.34 | 1.37 | 1.97 | 10.55 | 11.13 | 20.00 | 21.49 | 0.38 | 0.49 | 2.72 | 3.23 | 84.46 | 30.97 | 76.56 | 80.93 | 0.37 | 0.41 |
| PAIF [25] | 2.43 | 2.51 | 13.16 | 8.38 | 44.41 | 38.21 | 0.37 | 0.46 | 2.30 | 3.14 | 7.26 | 7.50 | 24.16 | 24.63 | 0.36 | 0.54 | 3.03 | 3.30 | 19.55 | 15.68 | 81.41 | 83.13 | 0.38 | 0.36 |
| DDFM [64] | 2.31 | 2.07 | 15.06 | 13.33 | 38.51 | 40.83 | 0.51 | 0.59 | 2.21 | 3.11 | 10.78 | 9.06 | 31.91 | 34.77 | 0.57 | 0.85 | 4.38 | 3.94 | 27.95 | 13.31 | 61.54 | 66.77 | 0.52 | 0.51 |
| DRMF (Ours) | 3.13 | 3.13 | 25.12 | 25.12 | 54.06 | 54.06 | 0.72 | 0.72 | 3.68 | 3.68 | 13.81 | 13.81 | 57.70 | 57.70 | 0.69 | 0.69 | 3.36 | 3.36 | 25.72 | 25.72 | 87.72 | 87.72 | 0.63 | 0.63 |

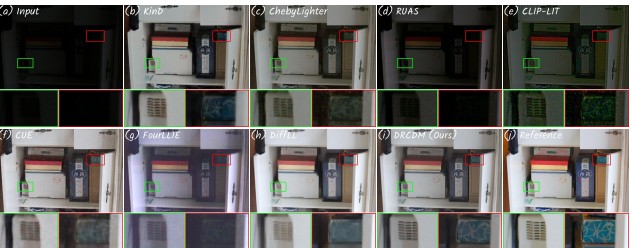

**Figure 6: A typical example of low-light enhancement.**

fusion results distinctly present scene information (*e.g.*, abundant textures and prominent objectives in IVIF), even amidst complex image degradation. Additionally, only DRMF can synthesize the cleanest and sharpest fused images in MIF, owing to the exceptional degradation elimination capability of DRCDM. In contrast, other alternatives suffer from the harsh effects of composite degradation, leading to decreased fusion performance. Although employing cascaded pre-enhancement methods can mitigate these effects, it usually amplifies deficiencies in specific tasks, leading to unsatisfactory performance. Table 2 further substantiates the remarkable performance of DRMF in degraded fusion scenarios. Particularly, DRMF exploits the composable property of diffusion models to organically integrate enhancement into fusion tasks, avoiding incompatibility between various tasks and enabling impressive performance. In summary, both qualitative and quantitative experiments on IVIF and MIF demonstrate the superiority of DRMF in inhibiting degradation and aggregating complementary information.

## 4.4 Extension Experiment

**Low-light image enhancement.** We verify the effectiveness of our DRCDM on low-light image enhancement (LLIE), which is a prominent image restoration task. DRCDM is compared with several mainstream algorithms, including KinD [61], ChebyLighter [31], RUAS [24], CLIP-LIT [19], CUE [66], FourLLIE [46], and DiffLL [12]. A typical example is shown in Figure 6. It is evident that only KinD, CUE, DiffLL, and DRCDM can faithfully render the scene with natural brightness. Remarkably, DRCDM with powerful generative ability can mine information in the darkness, as highlighted by rectangular boxes. The quantitative assessments, involving PSNR, SSIM, MUSIQ [14], and LIQE [59], are presented in Table 3. The best

**Table 3: Quantitative comparison results on LOL dataset.**

| | KinD | ChebyL. | RUAS | CLIP-LIT | CUE | FourLLIE | DiffLL | DRCDM |
|---|---|---|---|---|---|---|---|---|
| PSNR | 19.39 | 22.40 | 12.77 | 14.13 | 24.57 | 20.81 | 30.76 | 26.13 |
| SSIM | 0.78 | 0.75 | 0.44 | 0.49 | 0.77 | 0.68 | 0.84 | 0.83 |
| MUSIQ | 68.88 | 65.55 | 47.63 | 60.12 | 61.90 | 57.92 | 67.42 | 72.66 |
| LIQE | 2.84 | 3.47 | 2.86 | 2.12 | 2.51 | 2.55 | 2.78 | 3.90 |

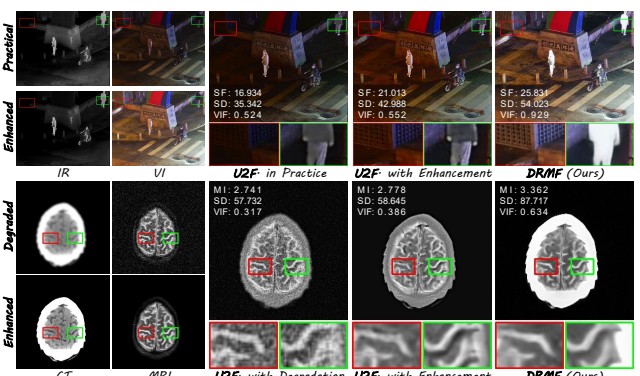

**Figure 7: Pre-enhancement for existing approaches.**

MUSIQ and LIQE indicate that DRCDM achieves the best perceptual quality, while elevated PSNR and SSIM imply that our results closely match reference images. These demonstrate the capability of DRCDM in addressing challenging image degradations. Qualitative and quantitative experiments on the LLIE task collectively demonstrate the practicability of DRCDM in addressing image degradation challenges. Notably, DiffLL not only achieves optimal SSIM and PSNR, but also has considerable visual effects, thus we adopt it as the LLIE preprocessing method for other fusion methods.

**Pre-enhancement for other fusion methods.** Our DRCDM can directly enhance source images as preprocessing for other fusion approaches. Figure 7 displays some representative examples. DRCDM yields high-quality source images for both actual and degraded scenarios. When existing fusion methods take enhanced source images as input, both qualitative and quantitative results show significant improvements. For instance, the background (*e.g.*, fences) in VI images and structure in MRI images are presented more clearly, and the SD and VIF metric are increased remarkably.

Table 4: Object detection on LLVIP. mAP is the average of all APs at various IoU thresholds, from 0.5 to 0.95 in steps of 0.05.

| Method | Practical scenarios | | | | | | | | Challenging scenarios | | | | | | | |
|---|---|---|---|---|---|---|---|---|---|---|---|---|---|---|---|---|
| | Prec. | | Recall | | AP@.50 | | mAP | | Prec. | | Recall | | AP@.50 | | mAP | |
| | Orig. | En. | Orig. | En. | Orig. | En. | Orig. | En. | Orig. | En. | Orig. | En. | Orig. | En. | Orig. | En. |
| **IR** | 0.857 | 0.857 | *0.764* | 0.764 | *0.834* | 0.834 | *0.499* | *0.499* | 0.737 | 0.649 | 0.652 | 0.595 | 0.705 | 0.658 | 0.359 | 0.306 |
| **VI** | 0.820 | 0.815 | 0.578 | 0.621 | 0.664 | 0.703 | 0.345 | 0.371 | 0.808 | 0.865 | 0.532 | 0.674 | 0.586 | 0.748 | 0.316 | 0.402 |
| **DeF.** [18] | 0.854 | **0.903** | 0.723 | 0.726 | 0.797 | 0.827 | 0.449 | 0.482 | 0.824 | 0.827 | 0.715 | 0.735 | 0.796 | 0.802 | 0.439 | 0.454 |
| **TarDAL** [20] | 0.870 | 0.874 | 0.682 | 0.740 | 0.803 | 0.831 | 0.456 | 0.483 | 0.841 | 0.868 | 0.715 | 0.740 | 0.801 | 0.826 | 0.453 | 0.457 |
| **U2F.** [52] | *0.886* | 0.863 | 0.666 | 0.710 | 0.775 | 0.803 | 0.432 | 0.462 | **0.882** | 0.851 | 0.717 | 0.715 | 0.807 | 0.787 | *0.459* | 0.444 |
| **DIVF.** [40] | 0.863 | 0.837 | 0.719 | **0.791** | 0.810 | 0.841 | 0.457 | 0.487 | 0.842 | 0.812 | 0.711 | 0.684 | 0.794 | 0.762 | 0.442 | 0.424 |
| **LRRNet** [17] | 0.867 | 0.881 | 0.663 | 0.690 | 0.753 | 0.781 | 0.421 | 0.446 | 0.866 | 0.862 | 0.714 | 0.715 | 0.794 | 0.790 | 0.449 | 0.445 |
| **MURF** [54] | 0.872 | 0.889 | 0.697 | 0.739 | 0.785 | 0.817 | 0.451 | 0.483 | 0.817 | 0.870 | 0.650 | 0.702 | 0.745 | 0.788 | 0.421 | 0.440 |
| **PAIF** [25] | 0.843 | 0.876 | 0.733 | *0.771* | 0.816 | *0.844* | 0.458 | 0.489 | 0.860 | *0.873* | *0.770* | *0.757* | *0.833* | 0.837 | 0.467 | *0.488* |
| **DDFM** [64] | 0.846 | 0.863 | 0.689 | 0.721 | 0.781 | 0.811 | 0.437 | 0.475 | 0.844 | 0.872 | 0.693 | 0.702 | 0.780 | 0.791 | 0.437 | 0.446 |
| **DRMF** (Ours) | **0.894** | *0.894* | **0.770** | 0.770 | **0.850** | **0.850** | **0.507** | **0.507** | *0.876* | **0.876** | **0.780** | **0.780** | **0.845** | **0.845** | **0.500** | **0.500** |

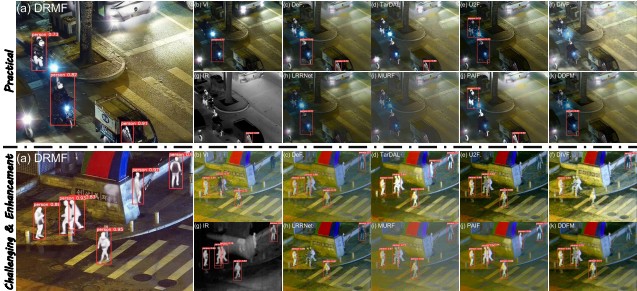

Figure 8: Visual comparison of object detection on LLVIP.

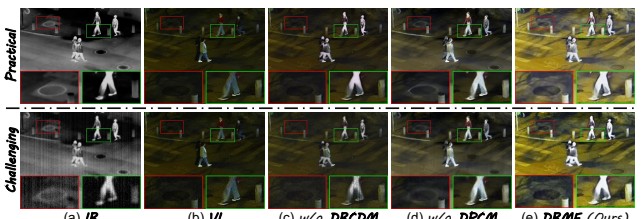

(a) IR    (b) VI    (c) w/o DRCDM    (d) w/o DPCM    (e) DRMF (Ours)

Figure 9: Visual comparison of ablation studies.

**Object detection.** Effective information enhancement and aggregation not only aid in visual perception, but also promote machine vision performance. Thus, we evaluate object detection performance on LLVIP to further reveal the superiority of DRMF, where the re-trained YOLO v8 [34] is employed as the detector. In Figure 8, DRMF successfully detects all pedestrians in both practical and challenging scenarios. In contrast, source images and other methods suffer from varying degrees of missed detection. The quantitative results in Table 4 show that DRMF balances precision and recall, thereby achieving better average precision (AP). In particular, the best mean AP (mAP) across various IoU thresholds implies that DRMF can adapt to different IoU settings.

### 4.5 Ablation Studies

**Degradation-robust conditional diffusion model (DRCDM).**
DRCDM harnesses the potent generative abilities of diffusion models to eliminate the effects of various degradations. As presented

Table 5: Quantitative comparison of ablation studies.

| | Practical scenarios | | | | Challenging scenarios | | | |
|---|---|---|---|---|---|---|---|---|
| | MI | SF | SD | VIF | MI | SF | SD | VIF |
| **w/o DRCDM** | **5.06** | *20.55* | 36.66 | *0.87* | *2.97* | *20.61* | 36.58 | *0.56* |
| **w/o DPCM** | 2.13 | 13.76 | *40.78* | 0.56 | 2.11 | 13.61 | *42.75* | 0.54 |
| **DRMF** | *3.82* | **25.83** | **54.02** | **0.93** | **3.13** | **25.12** | **54.06** | **0.72** |

in Figure 9(c), when replace DRCDM with a typical conditional diffusion model, providing the prior of reconstructing inputs, the results are faithful to source images. Specifically, the fused images retain common information in source images but fail to eliminate nasty degradations. As shown in Table 5, the higher MI means the fusion model without priors provided by DRCDM transfers more information from source images to fused images. Unfortunately, there is a noticeable degradation in textures and visual quality.

**Diffusion prior combination module (DPCM).** DPCM aims to adaptively integrate diffusion priors driven by various modalities, thus preserving crucial information. We perform an ablation study using a simple average weighting strategy instead of DPCM. As depicted in Figure 9(d), the fusion results tend to average the enhanced source images. Specifically, even with DRCDM providing robust priors, the fusion results still experience degradations due to the inability to adaptively aggregate relevant priors. The descent results in Table 5 also confirm this issue. In contrast, DRMF equipped with both DRCDM and DPCM can effectively cope with complex degradation and retain essential details and targets.

## 5 CONCLUSION

This work presents a degradation-robust multi-modal image fusion framework based on composable diffusion priors. The degradation-robust conditional diffusion models are devised to eliminate unfavorable degradations in source images and offer robust diffusion priors. Furthermore, we develop a diffusion prior combination module to adaptively aggregate diffusion priors driven by various modalities and progressively generate fused images. Experiments on IVIF and MIF demonstrate the superiority of our method in suppressing degradation and aggregating information during both practical and challenging scenarios.

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
