# OpenReview forum: "DRMF: Degradation-Robust Multi-Modal Image Fusion via Composable Diffusion Prior"
_acmmm.org/ACMMM/2024/Conference — MM2024 Poster_

### Official Review · Reviewer_qZoT · 2024-05-16

**Rating:** 5
**Confidence:** 4

**Summary:**

Based on the noise process of the diffusion process, the article believes that composable diffusion prior is helpful for multivariate image fusion and is robust to degradation. The article uses this to design a two-stage training Diffusion model. In the first stage, the diffusion model of each modality is trained. In the second stage, a diffusion priori combination module is trained separately to perform a weighted summation process of the noise of different modes. Advanced results on multiple datasets demonstrate its effectiveness.

**Strengths:**

1. The article introduces a diffusion model to provide a good degradation-robust single-modal prior.
2. Use the single-modal prior input trained in stage 1 to dynamically weight the information of different modalities.
3. Some ablations show the effectiveness of the proposed method. And the detection task shows its potential of applicaition.

**Limitations:**

1. What the shape of $\gamma_t^i$ is? $\mathbb R^{H\times W\times C}$ or just a scalar?
2. According to Eq. (10), is there an arrow from $x_t^f$ to Diffusion Priori Combination Module?
3. A major concern is that: after the noise prediction network $\epsilon_\theta^{i}$ are trained, the diffusion process $x_t$ in stage 1 and in the stage 2 are different. In stage 1, the $x_t$ is noisy (and degraded) single-modal visible image (or infared image), but in stage two, the $x_t$ is changed to fused image $x_t^f$. There exists an domain shift for the noise prediction network between stage one and two. The authors should consider it and explain why this does not affect the fusion performance. Moreover, the U-Net $\varphi$ only predict weights for combinate the predicted noise $\epsilon_\theta^i(x_t^f, x_c^i, t)$, it still can not mitigate the domain shift.

If the authors solves the concern 3., I am willing to rise my rating.

---

After rebuttal:

1. This explanation is quite useful, although domain shift still exists (even though the authors did not deny this), but the experimental figure shows it still works. Anyway, I rise the rating to "Weak Accept". Authors should add this discussion to the manuscript (This is important).

**Suitability:**

3

---

### Official Review · Reviewer_gqyW · 2024-05-24

**Rating:** 5
**Confidence:** 2

**Summary:**

In this work, authors propose Degradation-Robust Multi-modality image Fusion (DRMF), which utilizes the robust generative properties of diffusion models to address various degradations in image fusion. They pre-train multiple degradation-robust conditional diffusion models for different modalities to handle degradations effectively. Following this, they develop a diffusion priori combination module to merge the generative priors from these pre-trained uni-modal models, facilitating efficient multi-modal image fusion.

**Strengths:**

1.	The writing of the paper is good. The authors clearly identify the problems in existing MMIF methods and explain how their work addresses and solves these issues.
2.	Using the diffusion model to address various degradations in image fusion is intuitively feasible.
3.	The authors conducted comprehensive experiments to verify the effectiveness.

**Limitations:**

The model's performance appears to be good. However, the authors need to focus on discussing computational cost. Will the use of diffusion models and related designs lead to increased computational costs? How can the trade-off between performance and computational cost be achieved?

**Suitability:**

3

---

### Official Review · Reviewer_4S25 · 2024-05-24

**Rating:** 5
**Confidence:** 3

**Summary:**

This paper proposed Degradation-Robust Multi-modality image Fusion (DRMF), which leverages generative diffusion models to counteract various degradations during image fusion. By pre-training multiple degradation-robust conditional diffusion models for different modalities and integrating their generative priors, DRMF excels in infrared-visible and medical image fusion, even under complex degradations.

**Strengths:**

1. Extensive experiments demonstrate the outstanding performance of the proposed model.
2. The figures are aesthetically pleasing and effectively convey the motivation and process of the proposed method.
3. This proposed Degradation-Robust Multi-modality image Fusion (DRMF) is a novel way to integrate generative priors from pre-trained uni-modal models, enabling effective multi-modal image fusion.

**Limitations:**

1. Pre-training multiple degradation-robust conditional diffusion models may be computationally intensive and time-consuming.
2. Integrating generative priors from pre-trained uni-modal models might introduce complexity to ensure effective fusion.
3. There could be challenges in maintaining the balance between different modalities.

**Suitability:**

3

---

### Official Review · Reviewer_i7Uj · 2024-06-03

**Rating:** 3
**Confidence:** 2

**Summary:**

Existing multimodal fusion algorithms are unable to solve the low-quality degraded image fusion, based on this, this paper proposes a degradation-robust multimodal image fusion method, which utilizes the powerful generative properties of diffusion model to offset various degradations in the image fusion process.

**Strengths:**

In this paper, a new multimodal image fusion framework based on diffusion models is proposed, which utilizes the powerful generative properties of diffusion models to mitigate various complex degradations in the source images. Also in this paper, a diffusion prior combination module is developed to aggregate generative diffusion priors from different modalities, effectively exploiting the complementary nature of diffusion models driven by various modalities and degradations.

**Limitations:**

The introduction of infrared-visible image fusion and medical image fusion as two side-by-side tasks is not considered reasonable, and image fusion in industrial field and medical image fusion can be treated as a kind of side-by-side task.

The ablation experiments in this paper are not sufficient, and detailed ablation experiments should be conducted to verify the necessity and role of the designed module.

**Suitability:**

2

---

### Meta-Review · Area_Chair_44Hw · 2024-07-02

**Recommendation:** Accept (Poster)
**Confidence:** 5

**Metareview:**

The submission presents a novel and promising approach to multimodal image fusion that addresses significant challenges in the field. While there are concerns regarding the computational demands and the need for more rigorous experimental validations, the strengths of the paper—particularly its innovative approach and strong experimental results—suggest that it would be a valuable contribution to the conference. I recommend an Accept, encouraging the authors to address the highlighted concerns regarding computational efficiency and further experimental validations in future revisions or extensions of this work.